# Research on Mechanical Properties and Damage Evolution of Pultruded Sheet for Wind Turbine Blades

**DOI:** 10.3390/ma15165719

**Published:** 2022-08-19

**Authors:** Ying He, Yuanbo Wang, Hao Zhou, Chang Li, Leian Zhang, Yuhuan Zhang

**Affiliations:** 1Shandong University of Technology, Zibo 255049, China; 2Sinoma Technology (Funing) Wind Power Blade Co., Ltd., Yancheng 224400, China

**Keywords:** wind power blade, pultruded sheet, composite material, static tensile test, infrared thermal imaging method, testing machine

## Abstract

In order to explore the mechanical properties, failure mode, and damage evolution process of pultruded sheets for wind turbine blades, a tensile testing machine for pultruded sheets for wind turbine blades was built, and the hydraulic system, mechanical structure, and control scheme of the testing machine were designed. The feasibility of the mechanical structure was verified by numerical simulation, and the control system was simulated by MATLAB software. Then, based on the built testing machine, the static tensile test of the pultruded sheet was carried out to study the mechanical properties and failure mode of the pultruded sheet. Finally, an infrared thermal imager was used to monitor the temperature change on the surface of the test piece, and the temperature change law and damage evolution process of the test piece during the whole process were studied. The results show that the design scheme of the testing machine was accurate and feasible. The maximum stress occurred in the beam after loading the support, the maximum stress was 280.18 MPa, and the maximum displacement was 0.665 mm, which did not exceed its structural stress-strain limit. At the same time, the control system met the test requirements and had a good follow-up control effect. The failure load of the pultruded sheet was 800 kN. The failure deformation form included three stages of elasticity, yield, and fracture, and the finite element analysis data were in good agreement with the test results. The failure modes were fiber breakage, delamination, and interfacial debonding. The surface temperature of the specimen first decreased linearly, and then continued to increase. The strain and temperature trend were consistent with time.

## 1. Introduction

According to the outline of the 14th Five-Year Plan and the proposal for 2035, China will accelerate the development of new energy. Compared with traditional carbon-based energy, wind energy has huge advantages in terms of sustainable development. In order to reduce the cost of electricity, the size and power of wind turbines are increasing. The average power rating of wind turbines has increased from 0.1 MW in 1985 to an average of 10 MW today [1,2]. It is expected that the power of the wind turbine will reach 20 MW in the future, and the diameter of the impeller will be about 200 m [3]. The traditional manual lay-up technology can no longer meet the increasing blade size, and the plates produced by the pultrusion process with high forming quality, high production efficiency and low production cost can be used for key structural parts such as the main beam of wind turbine blades [4]. Due to technical risks and high costs, full-scale blade destructive testing still faces great challenges, so pultruded sheet testing has received more and more attention. Therefore, studying the failure mode and damage process of pultruded sheets representing blade design can be used to supplement full-scale blade testing and provide detailed mechanical performance parameters for the entire blade, which is of great engineering significance for reducing blade costs.

At present, a lot of work has been done on the research of pultruded glass fiber composites. A comprehensive presentation of pultruded fiber-reinforced polymers by Alessandro et al. [5] showed that pultruded fiber reinforcements had undergone extensive experimental and numerical studies to evaluate their performance as structural components. A critical review of the state-of-the-art in numerical simulation for predicting mechanical behavior at the limit state of failure was also presented, and a variety of numerical simulation methods, including finite element methods, were introduced. Al-Saadi et al. [6,7] conducted tensile, compression and shear tests on pultruded glass fiber round tubes and square tubes to explore the mechanical properties and failure modes of the tubes and compared the numerical simulation with the test results to verify the accuracy of the test. The study explored the different specifications of materials produced by the pultrusion process, but not pultruded sheets. Madenic et al. [8] analyzed the bending mechanical properties and damage process of pultruded glass fiber composites by a three-point bending mechanical test. The theoretical solution and finite element analysis were carried out, and the obtained load-displacement curve was compared with the experimental results. It was found that the numerical simulation and theoretical solution results were consistent with the experimental values, but the article only analyzed the bending mechanical properties. Since the glass fiber is an anisotropic composite material, only the bending direction analysis is far from enough. Silva et al. [9] studied the flexural properties of dry and wet composites of polyester-based glass fiber reinforced plastic flat specimens cut from I-beam pultruded profiles, and found that the flexural modulus and strength of the material specimens wetted to the saturation level decreased. Paciornik et al. [10] explored the microstructure of pultruded glass fiber sheets and used an image analysis system to characterize the fiber size, spatial distribution and filler fraction of the sheets. The bending mechanical behavior of the sheet was determined, and it was found that in the pultrusion process, even if all parameters were controlled, the pultruded part still produced uneven fiber distribution and voids. In this paper, the microstructure of pultruded sheet was explored, but its mechanical properties and macroscopic damage evolution were not explored. Harizi et al. [11] used infrared thermal imaging technology to monitor the surface temperature changes of glass fiber composites under static load and stage load. Through experiments, it was found that infrared thermal imaging technology could monitor the evolution process of material damage, and the thermal imaging map could explore the material damage generated under large load conditions for analysis. Aniskevich et al. [12] designed a novel equipment to provide a good connection with a conventional existing pultrusion machine, allowing the production of pultruded pipes that involved cork and polyurethane foam preforms in a continuous way. The equipment had been developed and tested to obtain high flexural strength hybrid pultruded products. The above research shows that the technology of infrared thermal imaging monitoring of the surface temperature change of the test piece is very mature, and can be directly used for the study of the damage evolution of pultruded sheets, but the research on pultruded glass fiber sheets for wind turbine blades has not been involved. Its failure mode and damage evolution need to be specifically explored.

In order to solve the above problems, with the help of the conclusions of many scholars, this paper mainly studied the tensile mechanical properties and failure modes of pultruded glass fiber sheets, and explored the damage evolution process through infrared thermal imaging. First, a wind turbine blade pultrusion fiberglass sheet tensile testing machine was built. The finite element analysis of its structure was carried out and the control system was simulated by MATLAB software. Secondly, the static tensile test of the pultruded sheet was carried out using the built testing machine and compared with the numerical calculation results to study the mechanical properties and failure mode of the pultruded sheet. Finally, the damage evolution process was explored by using an infrared thermal imager to monitor the surface temperature of the pultruded sheet, which provided a reference for the mechanical properties of the entire wind turbine blade.

## 2. Test Principle

### 2.1. Pultruded Sheet Theory

Pultruded glass fiber reinforced polymer sheet, referred to as pultruded sheet for short, is made of pultruded fiberglass and epoxy resin through a pultrusion process [13]. In wind turbine blades, several pultruded sheets are laid to form the main beam structure [14,15,16,17]. The main beam structure is mainly determined by the mechanical properties, geometry and boundary conditions of the glass fiber. It is known that glass fiber is an orthotropic material, and each unit in the formula is an international standard unit. The constitutive relation in the plane stress state is:(1){σ}=Q{ε}

In the formula, σ is the stress, ε is the strain, and *Q* is the positive axis stiffness matrix.

After the positive axis stiffness matrix is obtained from the elastic constant of the glass fiber, its main direction is converted to the overall coordinate xyz, and the coordinate conversion matrix T and the corresponding relationship between stress and strain are obtained. Finally, we calculate the in-plane stiffness coefficient
(2)$$A_{ij}=\sum_{k=1}^{n}\left(\bar{Q_{ij}}\right)f\;(i=1,2,3;\;j=1,2,3)$$
where $\bar{Q}$ is the transformation stiffness matrix of Q:

The in-plane flexibility *S* is as follows:(3)$$S=A^{-1}$$
(4)$$S=\left[\begin{array}{ccc} 1/E_{1} & -v_{21}/E_{2} & 0\\ v_{21}/E_{2} & 1/E_{2} & 0\\ 0 & 0 & 1/G_{12} \end{array}\right]$$

The orthotropic equilibrium equation and deformation coordination equation are as follows
(5)$$\frac{\partial \sigma_{x}}{\partial x}+\frac{\partial \tau_{xy}}{\partial y}=0$$
(6)$$\frac{\partial \tau_{xy}}{\partial x}+\frac{\partial \tau_{y}}{\partial y}=0$$
(7)$$\frac{\partial^{2}\varepsilon_{x}}{\partial y^{2}}+\frac{\partial^{2}\varepsilon_{xy}}{\partial x^{2}}=\frac{\partial^{2}\gamma_{xy}}{\partial x\partial y}$$
where $\tau_{xy}$ is the shear stress between the x and y axes and $\gamma_{xy}$ is the shear strain.

The stress function F=(x,y) is introduced, and the relationship between the stress functions and $\sigma_{x}$, $\sigma_{y}$, and $\tau_{xy}$ is as follows:(8)$$\left\{ \begin{array}{l} \sigma_{x}=\frac{\partial^{2}F}{\partial y^{2}}\\ \sigma_{y}=\frac{\partial^{2}F}{\partial x^{2}}\\ \tau_{xy}=-\frac{\partial^{2}F}{\partial x\partial y} \end{array}\right.$$

The plane equation of the orthotropic blade composite material is obtained from Equations (1)–(8)
(9)$$S_{22}\frac{\partial^{4}F}{\partial x^{4}}+(2S_{12}+S_{66})\frac{\partial^{4}F}{\partial x^{2}\partial y^{2}}+S_{11}\frac{\partial^{4}F}{\partial y^{2}}=0$$
where $S_{11}=1/E_{1}$, $S_{12}=-v_{21}/E_{2}$, $S_{22}=1/E_{2}$, $S_{66}=1/G_{12}$.

The elastic modulus in the x-direction after regularization is as follows
(10)$$E_{x}=\frac{1}{tS_{11}}$$
where *t* is the total thickness of the plate.

According to the macroscopic treatment method, the fracture study of pultruded sheet can be regarded as a homogeneous anisotropic material. The fracture criterion of isotropic materials is extended to plate composite materials, and the stress field and displacement field near the origin are calculated by the elastic mechanics criterion
(11)$$\left\{ \begin{array}{l} \sigma_{x}=\frac{K}{\sqrt{2\pi r}}\cos\frac{\theta }{2}(1-\sin\frac{\theta }{2}\sin\frac{3\theta }{2})\\ \sigma_{y}=\frac{K}{\sqrt{2\pi r}}\cos\frac{\theta }{2}(1+\sin\frac{\theta }{2}\sin\frac{3\theta }{2})\\ \tau_{\mathrm{xy}}=\frac{K}{\sqrt{2\pi r}}\sin\frac{\theta }{2}\cdot \cos\frac{\theta }{2}\cdot \cos\frac{3\theta }{2}\\ u_{x}=\frac{K}{8G}\sqrt{\frac{2r}{\pi }}\left[\left(2\kappa -1\right)\cos\frac{\theta }{2}-\cos\frac{3\theta }{2}\right]\\ u_{y}=\frac{K}{8G}\sqrt{\frac{2r}{\pi }}\left[\left(2\kappa +1\right)\cos\frac{\theta }{2}-\sin\frac{3\theta }{2}\right] \end{array}\right.$$
where κ is the modulus of elasticity, *G* is the shear modulus, and *K* is the stress intensity factor.

The energy release rate *G* is the energy released per unit area of material by crack propagation. According to elastic fracture mechanics, its relationship with the strength factor *K* is described as
(12)$$\left\{ \begin{array}{l} G_{1}=\frac{1-v^{2}}{E}K^{2}\\ G_{2}=\frac{1}{E}K^{2} \end{array}\right.$$
where *G*_1_ is the plane strain energy release rate and *G*_2_ is the plane stress energy release rate.

### 2.2. Infrared Imaging Theory

Any object can emit infrared radiation. According to Stephen Boltzmann’s law in thermodynamics, the total energy radiated per unit area of an object’s surface in a unit time is proportional to the fourth power of the thermodynamic temperature of the object itself [18,19]. The formula is as follows:(13)$$j^{*}=\varepsilon \sigma T^{4}$$

Among them, *j** is the radiance, *T* is the absolute temperature, *ε* is the emissivity of the black object, and *σ* is the Stefan constant.

Infrared thermal imaging detection technology is a non-destructive testing technology that converts invisible infrared radiation into visible images [20]. It is divided into active infrared thermal imaging and passive infrared thermal imaging [21,22]. Active infrared thermal imaging needs to excite the test piece by heating, so that the test piece forms a temperature difference [23]. Then, infrared imaging technology is used to test the specimen, and the defect is judged according to the difference of temperature [24,25]. In this test, it is assumed that a small change in temperature during the static tensile test of the testing machine will cause a large change in the radiation power, so there is no need to stimulate the curing of the test piece. The passive thermal imager is used for detection, which is presented in the form of a thermal image, and the detection is realized by comparison.

In addition, heat is transferred from a hotter part of an object to a cooler part, or from a hotter object to another cooler object in contact with it. This heat transfer process is called heat conduction [26]. The heat flow in the pultruded sheet can be determined according to Fourier’s theorem [27]:(14)q(r,t)=−λ∇T(r,t)

In the formula, *q*(*r*, *t*) is the heat flow per unit area per unit time in the direction of decreasing temperature gradient; *λ* is the thermal conductivity of wind turbine blades, W/(m∙K); and *T*(*r*, *t*) is the spatial and temporal distribution of the temperature in the wind turbine blade.

Equation (14) reveals the relationship between heat flow and temperature. The internal relationship of the temperature field in the spatiotemporal domain is usually described by the heat conduction differential Equation (15)
(15)$$\nabla T(r,t)+\frac{q_{v}}{\lambda }=\frac{\rho c}{\lambda }\nabla T(r,t)$$
where *q_v_* is the loading heat source term; *ρ* is the density of the wind turbine blade, kg/m^3^; and *c* is the specific heat capacity of the wind turbine blade, J/(kg∙°C).

## 3. Overall Test Plan and Simulation

### 3.1. Test Machine Structure Scheme

The structure of the pultruded sheet testing machine is composed of a hydraulic system and a loading mechanism, including loading brackets, fixtures, force sensors and oil cylinders, which can perform horizontal tension and compression tests. At the same time, in order to offset the radial force generated during loading, the testing machine is equipped with a ball head mechanism on the front beam and the rear beam. The pultruded sheet is connected to the fixture by bolts, and the bolts are subjected to axial load to fasten the specimen in the fixture. The schematic diagram of the mechanical structure is shown in Figure 1.

When testing in the horizontal direction, the hydraulic cylinder installed on the front beam exerts an axial force on the component through the cylinder tie rod, which is then transmitted to the sensor installed on the rear beam. From this, it can be seen that the rear beam is the main component that bears the load. The design parameters are shown in Table 1. The force sensor of the front beam converts the electricity into electrical signals through the transmitter, and feeds them back to the host computer.

### 3.2. Structural Finite Element Analysis

The loading bracket of the testing machine was numerically simulated in Ansys workbench, and its finite element model was shown in Figure 2. The pre-processing steps for static analysis using the finite element method included assigning material properties, dividing meshes, selecting elements, applying constraints and loads, etc. [28,29,30]. The holes and grooves that generated stress concentration and did not affect the structural strength were simplified. The loading bracket material was Q345 structural steel, and the material parameters were shown in Table 2. The Solid185 element was used to construct three-dimensional solid structures. Due to its large deformation and large strain capacity [5], this element was used. Since the mesh division had a great influence on the results of the finite element analysis [31], the mesh of the sensor directly bearing the force on the rear beam was refined. The element type, total number of nodes, and mesh size after meshing are shown in Table 2. The front beam and the rear beam of the testing machine have four supports respectively, which are connected with ground bolts to prevent the testing machine from overturning, and the contact surface with the ground is set as a restraint surface. When an external load is applied, the pultruded sheet produces a load towards the inside of the contacting load bracket.

Grid independence verification was performed first. In Ansys workbench, by setting the target value, the software automatically calculated many times to determine the grid-independent solution. Inserted Convergence in the resulting Equivalent Stress. Since it was simpler to load the scaffold model, we inserted a delta of 0%. After each solution, the software automatically re-divided the mesh in the area that needs to be refined, calculated the corresponding equivalent stress, and compared it with the previous result until the difference between the results met the set requirements. Then, numerical simulation was carried out. After 4 calculations, the maximum stress of the structure was calculated to be 280.18 MPa, which was located at the contact point between the pressure sensor and the front beam. In addition, the stress at the bottom connection reached 249.05 MPa, as shown in Figure 3. The result was not the allowable stress of the metamaterial, which could meet the testing requirements. The safety of the structure was demonstrated.

At the same time, the displacement results of the loading bracket and other components of the testing machine obtained by running the example are shown in Figure 4. The maximum deformation is 0.665 mm, and the combined deformations of the front-end fixture, rear-end fixture, and bolt of the specimen are 1.381 mm, 2.101 mm, and 0.4485 mm, respectively, all of which meet the design requirements.

### 3.3. Test Control Scheme and Simulation

The design scheme of the static tensile test of pultruded sheet is shown in Figure 5. The test consists of a tensile testing machine and an infrared thermal imaging test system. The control system of the testing machine is composed of the actuator, the lower computer, the upper computer, and the data acquisition system. The host computer is written in LabVIEW language, and the overall control of the platform is carried out through the PC terminal. The upper computer has the functions of programming the control algorithm, collecting data, storing and displaying the strain curve. The lower computer is composed of PLC and RMC in parallel, in which PLC obtains signals such as force sensor, oil temperature sensor and liquid level sensor. RMC control electro-hydraulic servo valve has the advantages of fast response speed, high control precision and good dynamic performance. The execution structure adopts the MOOG servo valve, which has the advantages of high control precision and fast dynamic response. It can perform high-frequency reversing action and drive sub-components to perform fatigue tests through the pulling and pressing action of the oil cylinder. At the same time, the infrared thermal imager observes the workpiece in real time and collects data, and transmits the surface temperature data and thermal image of the test piece to the PC.

During the test, the host computer sends a control signal to the controller and then controls the servo valve to make the oil circuit on and off and change direction, so as to load the plate with tension. The PLC collects the signal of the force sensor and feeds it back to the host computer. The signal is processed by the control algorithm in the host computer to realize the follow-up control of the actual load and the expected load. During the test, according to the IEC 61400-5 standard [32], it provides a data reference for the selection of the safety factor of blade material layout, as shown in the following Formula (16):(16)$$\gamma_{m}=\gamma_{m}{}_{0}\cdot \gamma_{m}{}_{1}\cdot \gamma_{m}{}_{2}\cdot \gamma_{m}{}_{3}\cdot \gamma_{m}{}_{4}^{}\cdot \gamma_{m}{}_{5}$$

In the formula, $\gamma_{m}{}_{0}$ is the safety factor of the base material; $\gamma_{m}{}_{1}$ is the environmental deterioration factor (irreversible influence); $\gamma_{m}{}_{2}$ is the temperature effect factor (reversible influence); $\gamma_{m}{}_{3}$ is the manufacturing effect factor; $\gamma_{m}{}_{4}^{}$ is the method calculation accuracy and verification factor; and $\gamma_{m}{}_{5}$ is the load characteristic factor.

Since the loading accuracy and loading frequency of the pultruded sheet are required to be high during the test process, the actual loading and the expected loading curve should ensure high synchronization, so that the slight difference between the two should be as close as possible, at 0. This means slight differences between the sinusoidal responses may also have a large impact on the test results. On the MATLAB/Simulink platform, the Sim Hydraulics module is used to build the simulation model of the control system of the testing machine, as shown in Figure 6. The figure shows the simulation of the hydraulic loading system control of the testing machine to achieve precise control of the axial load of the pultruded sheet, so as to ensure a good follow-up control effect between the actual axial load and the expected load of the pultruded sheet, and finally verify that the control system meets the test requirements. According to the test needs, the electro-hydraulic servo force control system is adopted, and the servo control is realized through the servo valve. The actuator of the system is a hydraulic cylinder, and one end of the hydraulic cylinder is connected with the fixture. The servo mechanism is a direct-acting servo valve, and the servo valve and the hydraulic cylinder form a valve-controlled cylinder structure. The hydraulic system is powered by a plunger pump, which is connected to the motor through a coupling. In order to reduce the flow pulsation caused by the plunger pump and ensure the smooth movement of the hydraulic cylinder, an accumulator is added to the oil circuit to reduce the corresponding vibration and noise. It should be noted that, since the components such as the cooler on the hydraulic station mainly play an auxiliary role in the hydraulic system, the simulation model is simplified to improve the efficiency of the control algorithm [33,34].

The simulation curve obtained based on the above model is shown in Figure 7, in which the actual response is the load of the specimen fed back by the force sensor, the expected response is the desired ideal sinusoidal waveform, and the error is the deviation between the actual response and the expected response. It can be seen from the figure that the actual response gradually follows the expected response after 1 s, and its followability can meet the design requirements of the algorithm, except for a small range of amplitude attenuation at the peak. Therefore, the control system meets the test requirements, and has a good follow-up effect.

## 4. Test Platform Construction and Test

### 4.1. Construction of Test Platform

Based on the design of the wind turbine blade pultrusion plate testing machine and the static stress analysis of the loading bracket, the construction of the testing machine was completed. The design parameters are shown in Table 3. The servo motor drives the hydraulic pump to convert the mechanical energy into hydraulic energy [35], and the hydraulic oil enters the hydraulic cylinder through the servo valve to work, as shown in Figure 8. The infrared thermal imager is UTi260B infrared thermal imager from Unitech. This thermal imager can monitor the temperature change on the surface of the specimen throughout the whole process. The acquisition frequency is 25 Hz, the pixel size is 320 × 240, and the thermal imaging sensitivity is less than 60 mK.

The static tensile test site of the pultruded sheet is shown in Figure 9. During the test, the ambient temperature was always 13.1°. The infrared thermal imager mainly monitored the surface temperature of the test piece. The infrared thermal imager was located on the outside of the test piece, and the lens was perpendicular to the test piece surface. During the experiment, a unified measurement standard was maintained, and the distance from the test piece was 30.00 cm. The strain measured after the ball head cancels the torque was the tensile and compressive strain. The fixture was fastened to the test piece by means of bolts. In order to obtain the strain of the pultruded sheet, the strain gauge was used in combination with the digital image system. The linear strain gauges A-4 to A-9 used for strain measurement were attached to the upper and lower surfaces of the expected failure area, and the rest of the strain gauges were at the transition position between the reinforced and unreinforced parts of the pultruded sheet, which could detect the detailed force data of the pultruded sheet.

### 4.2. Analysis of Results

After the test, the strain fed back by the force sensor and the digital image system and the temperature change of the surface of the pultruded sheet by the infrared thermal imager are shown in Figure 10. It was found that the strain of the specimen was consistent with the trend of the infrared temperature change during the static tensile test. The loading process went through elastic, plastic, and fracture stages. It can be seen from the curve that the change trend of the pultruded composite sheet in the static test is as follows:(1)From the beginning of loading to 20 s, the strain and surface temperature of the pultruded sheet decreased slightly, and the temperature decreased by about 1 °C. The strain drop in this process is due to the linear elastic deformation of the material. In the elastic deformation stage of the material, the material is subjected to less stress and will not yield, and the deformation can be quickly recovered. The reason for the temperature drop is the thermoelastic effect of the pultruded sheet without any damage to the surface. In the elastic deformation stage, the specimen is in the initial stage of tension, and the matrix stress is small, which is not enough to cause matrix damage.(2)After the specimen passed through the elastic stage, it continued to be subjected to the applied load. The strain and temperature values started to rise after reaching the minimum value, and the curve was nonlinear, which was manifested as a sharp rise at first and then a slow rise. At this time, the plate changed from linear elasticity to plastic deformation. When the specimen was stretched for 100 s, the infrared thermal image of the specimen changed rapidly, and a heat source with a significantly higher temperature appeared in the delamination defect. In this stage, the surface temperature of the specimen increased because the irreversible plastic deformation of the specimen began at this stage, and the mechanical work was converted into heat dissipation, which increased the temperature at the defect. Heat dissipation continued to increase cumulatively, making the heat source range larger with increasing load. When the temperature of the specimen reached the lowest point, the strain of the specimen presented a turning point. The stress value at this time was the yield value of the composite material, and the yield value of the specimen could be quickly determined by infrared thermal imaging technology.

In the plastic stage, the strain continued to increase, and no obvious change was found on the surface of the pultruded sheet. The number of damages at this stage increased, and the sound of fiber breaking was heard during the process, but the damaged part could not be seen with the naked eye. According to the preparation characteristics of the composite material, some damage phenomena such as resin cracking, delamination, and interfacial debonding had already occurred in the specimen during this tensile process.

(3)After a long period of plastic deformation, the specimen continued to be subjected to the applied load. After the specimen was loaded for 210 s, the specimen entered the fracture stage, and there was still no change on the surface. When the load reached 800 kN, the specimen broke rapidly.

After analyzing the variation trend of temperature and strain with time, the infrared thermal image obtained during the experiment was processed by the special infrared thermal image processing software that came with the infrared thermal imager. We selected the infrared thermal images at 20 s, 75 s, and 210 s with obvious changes during the test, as shown in Figure 11. From the infrared thermal images of Figure 11a–c, the temperature change trend and heat source distribution of the pultruded sheet composite material during the entire experimental process are obtained. Figure 11a shows that during the elastic deformation of the specimen, the surface temperature of the specimen is 0.97 °C lower than the environment, and the reason for the temperature drop is the thermoelastic effect of the pultruded sheet. Figure 11b shows that the specimen is in the plastic deformation period; at this time, the surface temperature of the specimen rises, and the temperature at the delamination damage position is higher than that in other places. Figure 11c shows that the specimen is fractured, and the surface temperature of the specimen reaches the highest at this time. The damage of the pultruded sheet in the static stretching process was verified through the elastic, plastic, and fracture stages by infrared thermal image.

Figure 12a shows the time domain diagram of the force sensor obtained. Each 100 kN increase in load during the loading process is a stage. The figure shows that the force increases linearly with time. After reaching 800 kN, the pultruded sheet broke and the load on the force sensor decreased sharply. At this time, the pultruded sheet still bores a large load. Over time, the pultruded sheet failed completely. Figure 12b is a comparison of experimental and numerically calculated strain-displacement data. During the numerical simulation calculation, the pultruded sheet is considered to be a uniform material, and the material parameters do not change with space. However, in the pultrusion process, the uniformity of the pultruded sheet cannot be strictly guaranteed [10], so the numerical simulation can only obtain a benchmark result [31], which cannot be strictly consistent with the test. Therefore, it can be seen that the linearization characteristic of the finite element model is stronger than that of the experimental data.

Figure 13 reveals the internal failure mechanism of the pultruded sheet. After the failure of the specimen, the surface is intact and there is no obvious fracture, which proves that the pultruded sheet composite material has a great damage tolerance. The specimen has no obvious signs before fracture, mainly because the fiber in the composite material is the main bearing capacity, and the layup angle is ±45°, which will prevent the crack propagation at the interface between the fiber and the matrix. Figure 13 shows fiber breakage along the 45° direction with interfacial debonding and delamination. According to the damage characteristics and on-site failure of composite materials, it is inferred that the fracture failure modes of pultruded sheets are fiber fracture, delamination and interfacial debonding.

## 5. Conclusions

In this paper, taking the mechanical properties and damage evolution of pultruded sheet composites for wind turbine blades as the research object, a pultruded sheet static tensile testing machine was built. Based on this, the static tensile test of the pultruded sheet composite material was carried out to explore the mechanical properties of the pultruded sheet for wind turbine blades. At the same time, the temperature change of the pultruded sheet composite material was monitored by infrared thermal imaging technology to explore the damage evolution of the pultruded sheet. The main conclusions are as follows:(1)Through the finite element analysis of the loading bracket of the testing machine, it was obtained that the beam after the loading bracket was the key part to bear the load. When the design load of the loading bracket was applied, the maximum stress of the loading bracket was 280.18 MPa, and the maximum deformation was 0.665 mm, which was lower than the yield limit of the Q345 material and met the requirements of the test machine. Using the MATLAB/Simulink platform to simulate the electro-hydraulic servo force control system, it was verified that the control system met the test requirements and had a good follow-up control effect.(2)When the pultruded sheet composite was subjected to static load, the failure load was 800 kN, and it would go through elastic, plastic, and fracture stages during the loading process. In the elastic deformation stage, the material was under less stress and could quickly recover from deformation. In the plastic stage, the strain first rose sharply and then rose slowly, which was irreversible at this stage. Finally, in the fracture stage, the pultruded sheet composites had damages such as fiber fracture, delamination, and debonding under static load. During the test, there was internal damage but no obvious fracture, which proved that the specimen had a large damage tolerance. The full-scale test of wind turbine blades was supplemented by exploring the mechanical properties of pultruded sheets to provide data reference for the mechanical performance parameters of the entire blade.(3)When the damage mode of the pultruded sheet was detected by infrared thermal imaging, the surface temperature of the specimen in the elastic stage decreased by about 1 °C. As the load increased, the specimen went through the plastic stage and the fracture stage, and the surface temperature gradually increased. The temperature variation trend was consistent with the strain, and its variation range was 5 °C.(4)Infrared thermal imaging technology can analyze the damage evolution process of layered composites under load from the monitored infrared thermal images. When the temperature of the surface of the test piece is monitored by an infrared thermal imager, the yield limit of the test piece can be determined, which is beneficial to quickly determine the performance parameters of the test piece, and provides a simple method for engineering applications.

## Figures and Tables

**Figure 1 materials-15-05719-f001:**
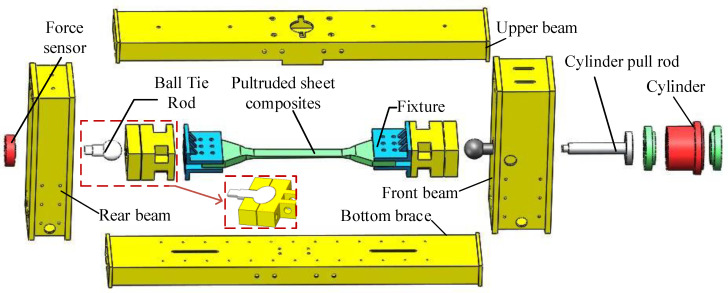
Test machine structure diagram.

**Figure 2 materials-15-05719-f002:**
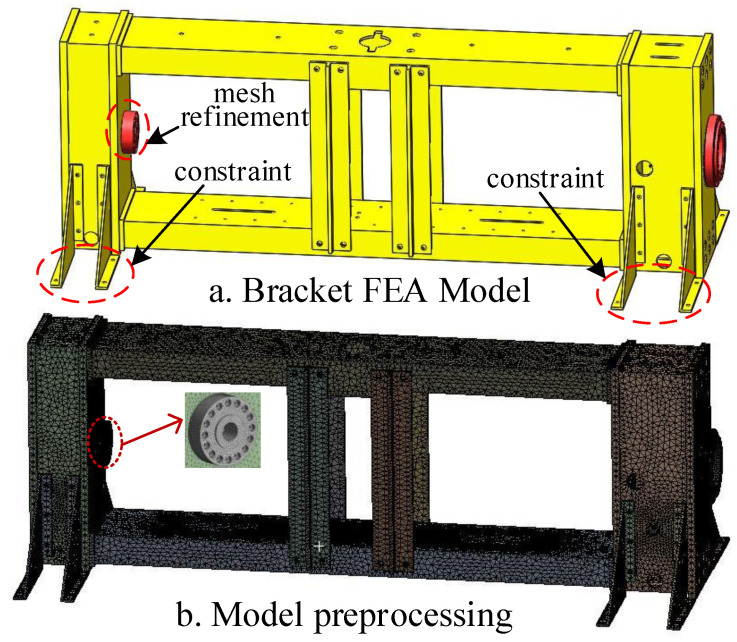
Finite element modeling and pre-processing: (**a**) bracket FEA model; (**b**) model preprocessing.

**Figure 3 materials-15-05719-f003:**
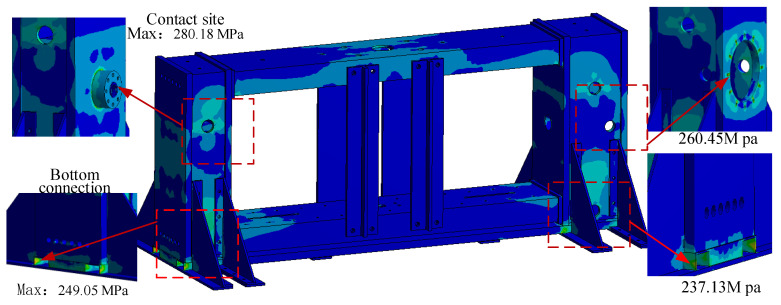
Loading bracket static stress analysis cloud diagram.

**Figure 4 materials-15-05719-f004:**
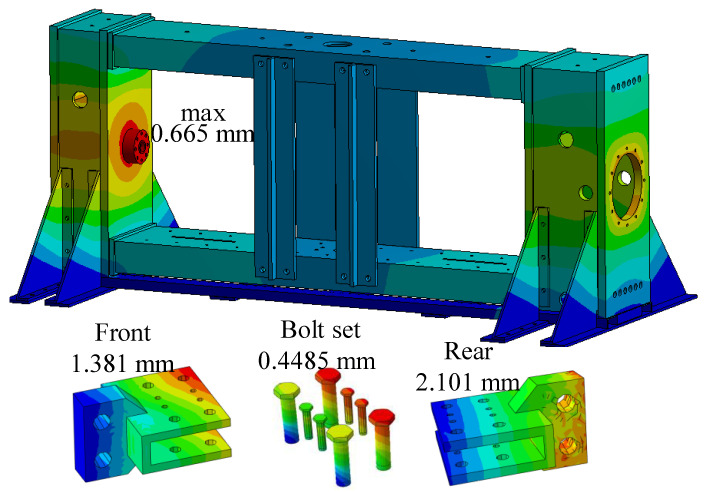
Displacement analysis cloud diagram of testing machine parts.

**Figure 5 materials-15-05719-f005:**
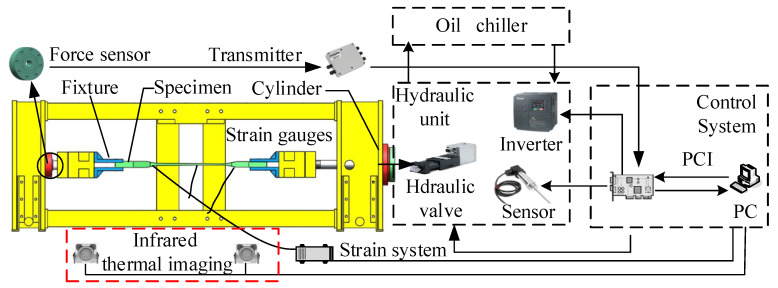
Experimental control scheme diagram.

**Figure 6 materials-15-05719-f006:**
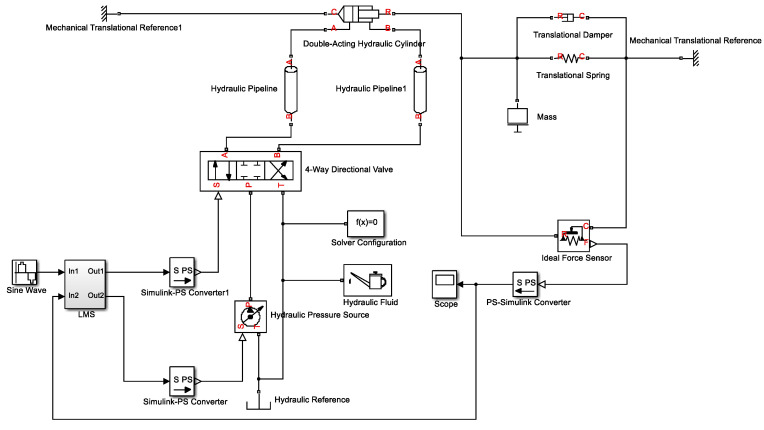
Simulation model of testing machine.

**Figure 7 materials-15-05719-f007:**
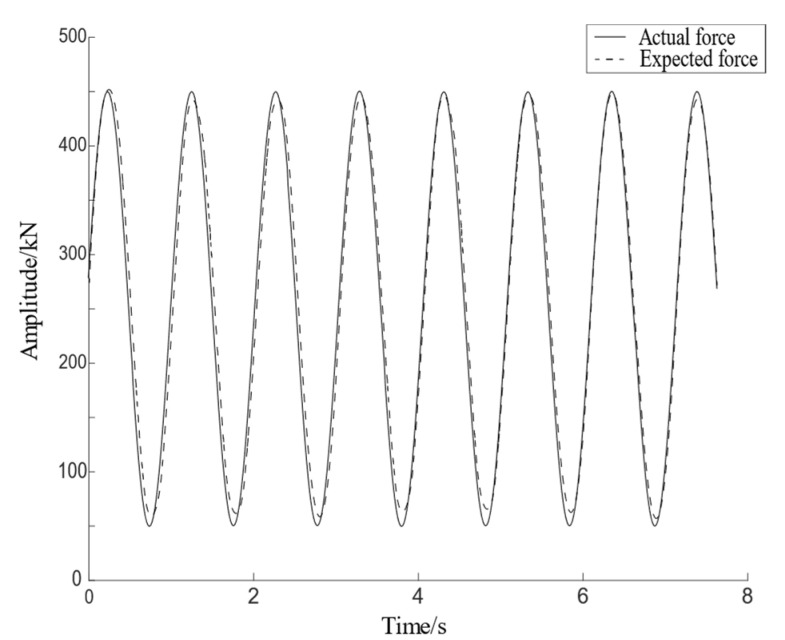
Simulation curve of testing machine.

**Figure 8 materials-15-05719-f008:**
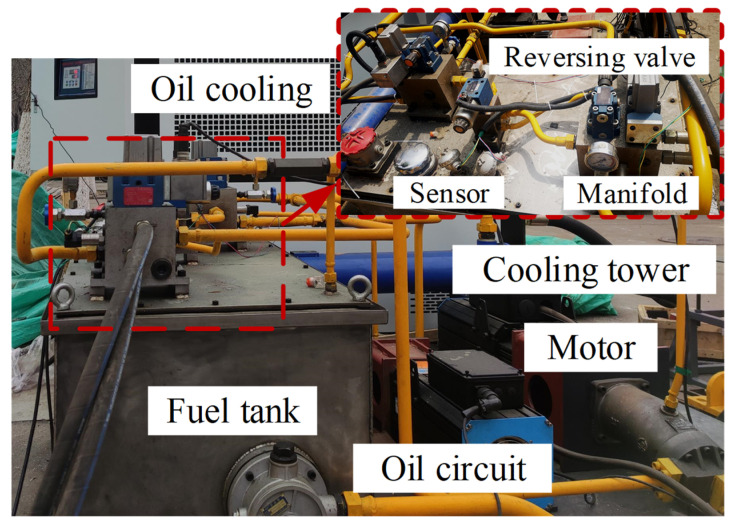
Testing machine hydraulic system.

**Figure 9 materials-15-05719-f009:**
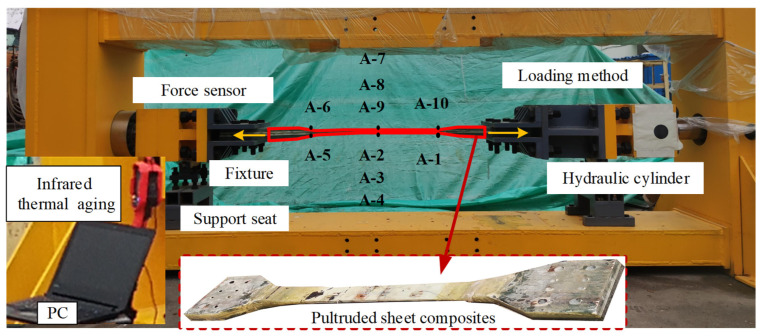
Static tensile test site of pultruded sheet.

**Figure 10 materials-15-05719-f010:**
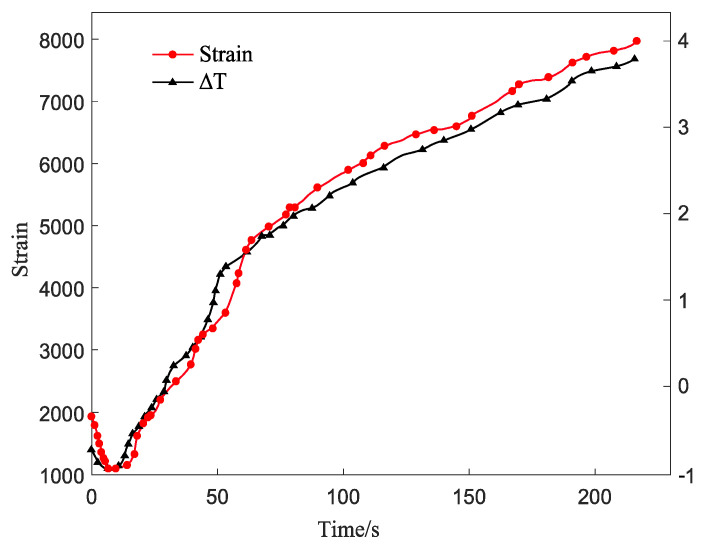
Specimen strain time domain diagram.

**Figure 11 materials-15-05719-f011:**
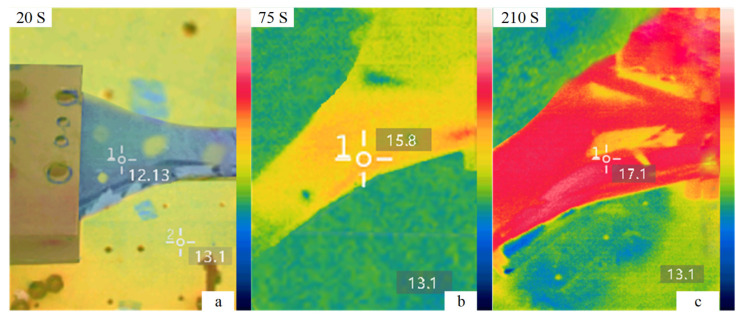
Infrared thermal images at different times. (**a**) Infrared thermal image at 20 s (**b**) Infrared thermal image at 75 s (**c**) Infrared thermal image at 210 s.

**Figure 12 materials-15-05719-f012:**
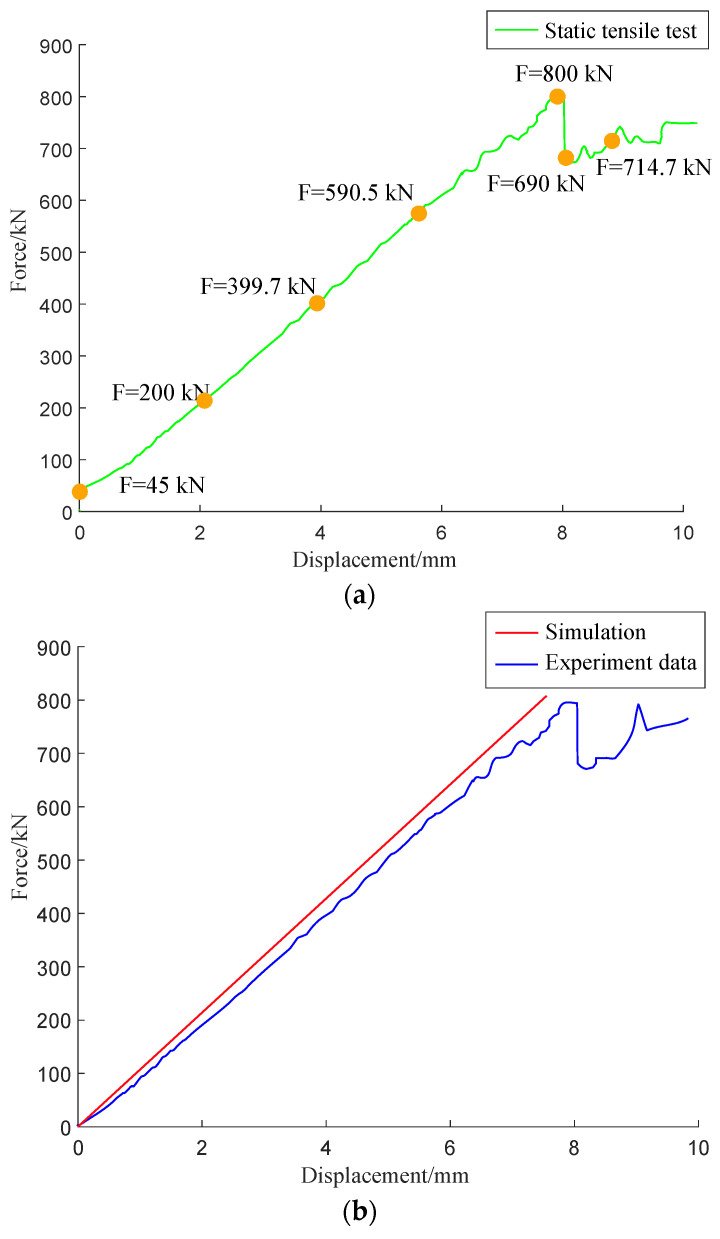
Force-displacement curve: (**a**) force sensor curve; (**b**) test-simulation curve.

**Figure 13 materials-15-05719-f013:**
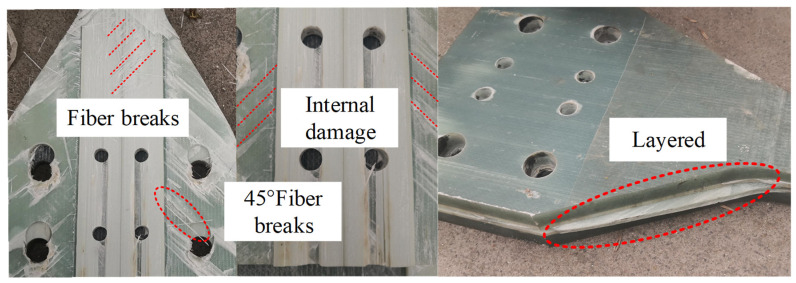
Failure area of pultruded sheet.

**Table 1 materials-15-05719-t001:** Design parameters.

Name	Parameter
Maximum specimen length (m)	2
Loading frequency (Hz)	1–3
Maximum load (kN)	1300
Motor power (kW)	36
Force sensor (kN)	150
Force sensor diameter (mm)	100
Sample thread	M36

**Table 2 materials-15-05719-t002:** Simulation parameters.

Name	Parameter
Gross weight (kg)	3912.3
Elastic modulus (GPa)	209
Poisson’s ratio	0.269
Yield strength (MPa)	345
Total number of nodes	270,136
Total number of units	142,078
Grid size (mm)	50
Front/rear beam grid size (mm)	15

**Table 3 materials-15-05719-t003:** Component test platform design parameters.

Name	Parameter
Specimen length (mm)	1760
Specimen thread	M36
Loading frequency (Hz)	1
Maximum load (kN)	1000
Servo valve	MOOG D661-G60KOAA4NSM2HA
Force sensor (kN)	150
Inverter	Invt CHF100A-022G-2
Motor speed (r/min)	1440
Motor power (kW)	36
Maximum working pressure (bar)	200
Pump maximum output pressure (bar)	31.5
Rotor brake feedback voltage (V)	24
Accumulator 1 volume (L)	2.8
Pump flow (L/min)	210
Air-cooled chiller	STSF-10
Rated throughput (T/h)	12

## Data Availability

No applicable.

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
