# Peer review of "Research on Mechanical Properties and Damage Evolution of Pultruded Sheet for Wind Turbine Blades"

_materials, 2022, doi:10.3390/ma15165719_

Round 1

Reviewer 1 Report

In this manuscript, the mechanical properties and damage evaluation of Pultruded sheet were investigated for wind turbine blades. The manuscript has the adequate novelty. Some questions and suggestions are provided before the acceptance of the presented manuscript:

-       The manuscript is not written in the format of the manuscript. It should follow the rules of material journal.

-       What is the usage of Figure 6 in this study?

-       Please present all Figures with higher quality.

-       In the conclusion make the possible summarizes as much as possible.

Author Response

  1. The manuscript is not written in the format of the manuscript. It should follow the rules of material journal.

We greatly appreciate the reviewer's suggestion that it is best to use the format given by the materials journal. After inquiries, the editorial board told the author that it is not necessary to format according to the format, and the editorial board will process it later. Authors will write in the materials journal format if you deem it necessary.

  1. What is the usage of Figure 6 in this study?

We are very grateful to the reviewers for their suggestions. Figure 6 is the simulation of the control system of the wind turbine pultrusion sheet testing machine, which is used to verify whether the system meets the test requirements and has a good follow-up control effect. Therefore, it has been revised based on the comments of the reviewers and the latest revision has been added in Figure 6.

  1. Please present all Figures with higher quality.

We greatly appreciate the reviewer's suggestion that it is best to make high-quality figures. Therefore, it has been revised based on the comments of the reviewers and the latest revisions have been added, in Figure 6 and Figure 11 et al.

  1. In the conclusion make the possible summarizes as much as possible.

We greatly appreciate the reviewer's suggestion for a better summary in the Conclusions. Therefore, according to the comments of the reviewers, it has been revised. The construction and control of the equipment in the text, the mechanical properties and failure modes, and the damage evolution process through infrared imaging are summarized. The latest revisions are added to the conclusions.

Reviewer 2 Report

The paper presents an interesting, very wide range of research for sheets of pultruded glass fiber. The destruction process was performed using infrared thermal imaging. An important stage of the work is the development of a machine for testing the tensile strength of pultruded sheets for wind farms. The obtained test results, such as: static tensile test of pultruded sheet and compared with the results of numerical calculations in order to investigate the mechanical properties of the pultruded sheet. The damage process was also determined using an infrared thermal imaging camera to monitor the surface temperature of the pultruded sheet metal, which provided some reference to the mechanical properties of the entire wind turbine blade.

Author Response

We are very grateful to the reviewers for their suggestions, the authors agree with the reviewers and have revised the abstract, introduction, conclusions and other parts of the article to make the article look more organized and polished the English grammar, again Thanks to the reviewers for acknowledging the article.

Reviewer 3 Report

Dear Authors,

Congratulations on your work, which is focused on a very interesting subject. As any other paper in this phase, there are some amendments to do, whose can improve the overall quality of your paper. Thus, I'm providing below some comments and suggestions, trying to collaborate by this way in improving your paper:

1. The Abstract doesn't clearly state the literature gap found, as well as the main motivation to develop this work. Thus, please clearly state the gap found in the literature in the Abstract, Introduction and Conclusions. The mains goals are also not clear in the Abstract. Please also chanfe "Mpa" by "MPa".

2. The novelty brought by your work is also not properly pointed out. Thus, please state clearly the novelty that your paper represents for the scientific community, stating as well if your contribution is exclusively scientific or if there was some practical motivation behind the development of your work. Any industrial application based on this work should also be pointed out.

3. The Literature Review is well done, and you are using  direct speech, describing briefly in what the work of previous Researchers has been focused on, methodology used and main results. However, the Literature Review is too short. Please consider to describe other studies, such as: doi: 10.1016/j.compositesb.2013.09.035; doi: 10.1016/j.compositesb.2013.09.035; doi: 10.1016/j.compositesb.2016.10.070; doi: 10.1016/j.compositesb.2017.12.019; doi: 10.1016/1359-8368(95)00015-1; doi: 10.1016/S0034-3617(06)71008-6; doi: 10.1590/0104-6632.20150322s00003181; doi: 10.1016/j.compstruct.2012.04.030.; doi: 10.1016/j.mcm.2006.02.006; doi: 10.1016/j.compositesb.2015.12.026.

4. The theoretical contribution of the paper is limited because no Methodology is pointed out. Please draw a schematic methodology showing your approach to the problem, helping others to follow your way.

5. The results are confusing, because there is two focus: the construction and control of the equipment, and the results of the tests and corresponding analysis. Please divede these two topics into different sections and describe each work separately.

6. No discussion is promoted, taking advantage of the Literature Review. How much your solutions are performing well regarding the initial solutions. What is the novelty? How can you quantify the gains using your approach?

7. Conclusions do not sound. They shold be divided into the two topics previously identified, highlighting the main achievements in each direction.

8. The number of references used is lower than expected.

Best whishes.

Kind regards. 

Author Response

  1. The Abstract doesn't clearly state the literature gap found, as well as the main motivation to develop this work. Thus, please clearly state the gap found in the literature in the Abstract, Introduction and Conclusions. The mains goals are also not clear in the Abstract. Please also chanfe "Mpa" by "MPa".

We greatly appreciate the reviewer's suggestion, and the authors agree with the reviewer's comments. The main objective of this paper is to investigate the mechanical properties, failure modes and damage evolution process of pultruded sheets for wind turbine blades. The author discusses the gap between this paper and other literatures in the introduction, and makes a specific study, that is, the technology of infrared thermal imaging monitoring of the surface temperature change of the test piece is very mature and can be directly used for the study of the damage evolution of the pultruded sheet, but The research on pultruded glass fiber sheets for wind turbine blades has not been covered, and its failure mode and damage evolution need to be specifically explored. Based on the reviewers' suggestions, the authors have made specific revisions in the Abstract, Introduction, and Conclusions, and the latest revisions have been added to the article.

  1. The novelty brought by your work is also not properly pointed out. Thus, please state clearly the novelty that your paper represents for the scientific community, stating as well if your contribution is exclusively scientific or if there was some practical motivation behind the development of your work. Any industrial application based on this work should also be pointed out.

We greatly appreciate the reviewer's suggestion, and the authors agree with the reviewer's comments. The innovation of the article is described in the introduction. The research in this article has certain practical applications. It is applied in the wind turbine testing industry through application, especially to provide data reference for the full-scale testing of wind turbines through the research in this paper. The authors have revised the Introduction, Abstract, etc., with the latest revisions added to the article.

  1. The Literature Review is well done, and you are using direct speech, describing briefly in what the work of previous Researchers has been focused on, methodology used and main results. However, the Literature Review is too short. Please consider to describe other studies, such as:

doi: 10.1016/j.compositesb.2013.09.035;

doi: 10.1016/j.compositesb.2013.09.035;

doi: 10.1016/j.compositesb.2016.10.070;

doi: 10.1016/j.compositesb.2017.12.019;

doi: 10.1016/1359-8368(95)00015-1;

doi: 10.1016/S0034-3617(06)71008-6;

doi: 10.1590/0104-6632.20150322s00003181;

doi: 10.1016/j.compstruct.2012.04.030.;

doi: 10.1016/j.mcm.2006.02.006;

doi: 10.1016/j.compositesb.2015.12.026.

We greatly appreciate the reviewer's suggestion, and the authors agree with the reviewer's comments. The authors conducted a study of other literature. And compared with the content studied in this paper. Based on the reviewers' comments, the authors have revised the Introduction, Abstract, etc., and the latest revisions have been added to the introduction of the article.

  1. The theoretical contribution of the paper is limited because no Methodology is pointed out. Please draw a schematic methodology showing your approach to the problem, helping others to follow your way.

We greatly appreciate the reviewer's suggestion. Based on the comments of the reviewers, the authors have drawn a schematic diagram of the abstract, as shown in Figure 1 below. However, the author believes that the process logic in the abstract of this article is clear, and it is not recommended to put the schematic diagram in the text. The authors will follow the reviewer's advice if the reviewer deems it necessary.

  1. The results are confusing, because there is two focus: the construction and control of the equipment, and the results of the tests and corresponding analysis. Please divede these two topics into different sections and describe each work separately.

We greatly appreciate the reviewer's suggestion, and the authors agree with the reviewer's comments. The focus of this paper is on the mechanical properties, failure modes and damage evolution of pultruded sheets. The device is built and controlled for simulation to be able to use the device for testing. This article provides a detailed analysis of the properties of pultruded sheets. The latest revisions have been added to the article test results.

  1. No discussion is promoted, taking advantage of the Literature Review. How much your solutions are performing well regarding the initial solutions. What is the novelty? How can you quantify the gains using your approach?

We are very grateful to the reviewers for their suggestions. The authors believe that there are two main innovations in this paper. First, a wind turbine pultruded sheet testing machine is constructed, through which the mechanical parameters and failure modes of the pultruded sheet can be tested. Second, the application of mature infrared imaging technology in experiments is an application innovation.

  1. Conclusions do not sound. They shold be divided into the two topics previously identified, highlighting the main achievements in each direction.

We greatly appreciate the reviewer's suggestion, and the authors agree with the reviewer's comments. The conclusions of this paper have been modified as suggested by the reviewers to the conclusions on the mechanical properties and failure modes of pultruded sheets and the conclusions on the damage evolution of pultruded sheets by infrared imaging. The latest revisions have been added to the conclusion of the article.

  1. The number of references used is lower than expected.

We greatly appreciate the reviewer's suggestion, and the authors agree with the reviewer's comments. The authors have cited additional references and the latest revisions have been added to the article references.

Reviewer 4 Report

The article is about mechanical properties and damage evolution of pultruded sheet for wind turbine blades. However, some issues must to be addressed:

  1. Abstract: Please start by expressing the aim of this paper, followed by the rest of the information. Typically, the abstract should provide a broad overview of the entire project, summarize the results, and present the implications of the research or what it adds to its field.
  2. Please avoid bulk citation like 1-3, 5-8 etc.
  3. The results are merely presented, not properly discussed. Please add explanations for the observed changes. Please give an extended discussion on the obtained results and correlate your findings with previous literature studies and prospective applications.
  4. Some captions are needed for figure 11.
  5. For the figures showing pictures is better to replace the text with some numbers and introduce captions.
  6. More analysis and interpretation of the results should be added for a clearer understanding of observed experimental phenomena.
  7. The authors must to provide some details about importance of the research and their applicability.
  8. Please rewrite the conclusions in a more quantitative form and enhance the clarity of the conclusion section in order to highlight the results obtained.
  9. General check-up and correction of the English language is suggested (i.e. figure 12). There are still some minor typos and grammatical errors.

The author needs to address the abovementioned points for the betterment of the manuscript.

Author Response

1.Abstract: Please start by expressing the aim of this paper, followed by the rest of the information. Typically, the abstract should provide a broad overview of the entire project, summarize the results, and present the implications of the research or what it adds to its field.

We greatly appreciate the reviewer's suggestion, and the authors agree with the reviewer's comments. And based on the comments of the reviewers, the latest revisions have been added to the abstract.

2.Please avoid bulk citation like 1-3, 5-8 etc.

We greatly appreciate the reviewer's suggestion, and the authors agree with the reviewer's comments. And based on the reviewers' comments, the article has been updated with the latest revisions, which have been added to the Introduction.

3.The results are merely presented, not properly discussed. Please add explanations for the observed changes. Please give an extended discussion on the obtained results and correlate your findings with previous literature studies and prospective applications.

We are very grateful for the suggestions of the reviewers, who feel that the results should only be presented without proper discussion. The authors agree with the reviewers to correlate the experimental phenomenon with other literature, discuss it, and apply it to the wind turbine blade testing industry.

4.Some captions are needed for figure 11.

We greatly appreciate the reviewer's suggestion, and the authors agree with the reviewer's comments. The author awakened a detailed explanation of the temperature change trend and heat source distribution of the pultruded sheet composite material obtained from the infrared thermal image in Figure 11 during the entire experimental process. And based on the comments of the reviewers, the latest revisions have been added to the article.

5.For the figures showing pictures is better to replace the text with some numbers and introduce captions.

We are very grateful for the reviewer's suggestion that for figures showing pictures it would be better to replace text with some figures and introduce a title. The author believes that labeling the text in the picture makes the various parts of the picture more intuitive. I'm very sorry, if the reviewer insists that it is better to use numbers, it can be suggested again, and the author will agree with the reviewer and make changes.

6.More analysis and interpretation of the results should be added for a clearer understanding of observed experimental phenomena.

We are very grateful for the suggestions of the reviewers, who feel that it corresponds to adding more analysis and interpretation of the results in order to gain a clearer picture of the observed experimental phenomena. The author agrees with the reviewer's suggestion, and adds more explanations to the test results, describing the experimental phenomenon of infrared imaging in more detail, and the latest revision has been added to the text test results.

7.The authors must to provide some details about importance of the research and their applicability.

We greatly appreciate the suggestion of the reviewers, who felt that the authors must provide some details about the significance of the study and its applicability. The author agrees with the reviewer's opinion that it is very necessary to study the mechanical properties and failure modes of pultruded sheets, which can reduce the full-scale damage test of wind turbine blades and reduce the cost of blades, which is of great engineering significance. And the research on the pultruded sheet for wind turbines is suitable for various types of test requirements of offshore wind turbine blades and road wind turbines. The latest revisions have been added to the article abstract.

8.Please rewrite the conclusions in a more quantitative form and enhance the clarity of the conclusion section in order to highlight the results obtained.

We are very grateful for the reviewer's suggestion that the conclusions should be rewritten in a more quantitative form and the clarity of the Conclusions section improved to highlight the results obtained. The author agrees with the reviewer's opinion. The latest conclusion is divided into two parts: the mechanical property parameters of pultruded sheet, the failure mode and the damage evolution process of infrared imaging, and the main achievements in each direction are expressed. The latest revision has been add to article conclusion.

9.General check-up and correction of the English language is suggested (i.e. figure 12). There are still some minor typos and grammatical errors.

We greatly appreciate the reviewer's suggestion. The reviewers feel that the English grammar of Figure 12 needs to be revised. Therefore, the author's manuscript has been revised by someone fluent in English and the latest revision has been added to Figure 12.

Round 2

Reviewer 3 Report

Dear Authors,

Thank you so much for addressing the Reviewers' comments and suggestions. However, there is still room for improvement. Thus, I'm providing some more suggestions allowing the improvement of your paper, as follows:

1. Introduction is now much better, but not strong enough for your paper. Please look for more papers related to pultruded composite structures to complete your Introduction. I think Prof. Luciano Feo has interesting work in that field, as well as Prof. Paulo Novo, MSc. Puria Esfandiari, Prof. João F. Silva, and so on. 

2. Please correct ".. AL saaid [ 5 6 ] et al....".

3. The way you are citing some previous works [5-11] is the best. If possible, please reinforce this part.

4. When describing the variables of the equations, please add the units for each variable, because there are differences between formulae express in SI or Royal systems of units.

5. In Figure 1, please explain how the fixture system is able to avoid samples' slipping. Figures 9 and 14 show some holes but this is not properly explained in the text.

6. In Table 1, please explain the units pointed out for "Force Sensor" (T). Is T Type? If so, it doesn't make sense.

7. Please clearly point out the software used to perform the simulations.

8. Please clearly point out the type of cells used in the simulation, and boundary conditions considered in Table 2.

9. Please avoid gross mistakes such as "Mpa" and "Gpa" (Table 2).

10.  In Figure 4, please insert a space between values and units. The same is valid for Figure 12.

11. After Figure 6, please explain item by item the corresponding function and the role developed in the system (need).

12. Regarding Figure 7, it is necessary to explain how a light difference between sinusoidal responses is so important.

13. In Figure 12 the results diverge as the load increase. Could this be avoided? Please explain how correct the simulations results.

14. In page 14, please correct "KN" to "kN".

15. The methodology/Approach/Strategy to develop this work doesn't exist. Thus, please try to establish and describe a strategy able to be transferred to other Researchers as Transferrable Knowledge.

16. No discussion of the results is presented. Please take advantage of the improvements need to be made in the Introduction to include results able to be compared at final of your work.

17. The novelty and gap found in the literature remain not clear in your work.

Good luck.

Kind regards.

Author Response

1. Introduction is now much better, but not strong enough for your paper. Please look for more papers related to pultruded composite structures to complete your Introduction. I think Prof. Luciano Feo has interesting work in that field, as well as Prof. Paulo Novo, MSc. Puria Esfandiari, Prof. João F. Silva, and so on.
We greatly appreciate the reviewer's suggestion. The reviewer believes that adding more relevant research in the introduction will make the paper more complete. According to the reviewer's suggestion, the author has added Professor Alessandro and others to the introduction to comprehensively describe the pultruded fiber-reinforced polymer, indicating that Pultruded fiber reinforcements have undergone extensive experimental and numerical studies to evaluate their performance as structural components, and a critical review of state-of-the-art numerical simulations for predicting mechanical behavior at failure limit states is presented. Various numerical simulation methods, including finite element method, are used. The latest revisions have been added to the introduction of the paper, again thanks to the reviewers for their suggestions.

2. Please correct "AL-saaid [5, 6] et al....".
We greatly appreciate the reviewer's suggestion. Based on the reviewer's suggestion, the author has revised Professor Alsaaid's research, and the latest revision has been added to the introduction of the paper. Thanks again for the reviewer's suggestion.

3. The way you are citing some previous works [5-11] is the best. If possible, please reinforce this part.
We greatly appreciate the reviewer's suggestion. The reviewer believes that the previous works [5-11] can explain more, making the article more complete. According to the reviewer's suggestion, the author has supplemented the research content of the previous works [5-11], and the latest revision has been added to the introduction of the paper. Thanks again for the reviewer's suggestion.

4. When describing the variables of the equations, please add the units for each variable, because there are differences between formulae express in SI or Royal systems of units.
We greatly appreciate the reviewer's suggestion. The reviewer believes that there will be differences between the use of SI units and imperial units. According to the reviewer's suggestion, the author adds that both use SI units in the content of the article. At the same time, the author consulted the editor and checked the format of the formula in other materials journals, but there was no relevant explanation. Based on this, the author only adds relevant instructions at the beginning, and the latest revision has been added to the first chapter of the paper. Thanks again for the reviewer's suggestion.

5. In Figure 1, please explain how the fixture system is able to avoid samples' slipping. Figures 9 and 14 show some holes but this is not properly explained in the text.
We greatly appreciate the reviewer's suggestion. The reviewer requested clarification on how the gripper system avoids slippage. According to the reviewer's suggestion, Zuola added that the extruded plate is connected to the fixture through bolts in the structural scheme of section 2.1 of the article, and the bolts are subjected to axial load to tighten the specimen in the fixture. The latest revision has been added to the paper. Figure 1, Figure 9 and Figure 14, thanks again to the reviewers for their suggestions.

6. In Table 1, please explain the units pointed out for "Force Sensor" (T). Is T Type? If so, it doesn't make sense.
We greatly appreciate the reviewer's suggestion. The force sensor (T) in Table 1 is the measurement range unit in China, which has been revised to the international standard unit kN, and the latest revision has been added to Table 1. Thanks again for the reviewer's suggestion.

7. Please clearly point out the software used to perform the simulations.
We greatly appreciate the reviewer's suggestion. The reviewer requested that the software used to perform the simulation be clearly indicated, and based on the reviewer's question, the authors have added a test machine loading bracket to the article for numerical simulation in Ansys workbench, the latest revision has been added to the article section 2.2. Thanks again for the reviewer's suggestion.

8. Please clearly point out the type of cells used in the simulation, and boundary conditions considered in Table 2.
We greatly appreciate the reviewer's suggestion. The reviewer asked to clearly indicate the type of element used in the simulation, and the boundary conditions considered in Table 2. According to the reviewer's question, the author added the Solid185 element to the article for the construction of three-dimensional solid structures with large deformations and large The strain capacity is determined by using this element, and the relevant literature is cited again to compare the element type and boundary conditions used in this paper. The latest revision has been added to Section 2.2 of the article. Thanks again for the reviewer's suggestion.

9. Please avoid gross mistakes such as "Mpa" and "Gpa" (Table 2).
We greatly appreciate the reviewer's suggestion. The authors modified "Mpa" to "MPa" and "Gpa" to "GPa" in Table 2. Thanks again for the reviewer's suggestion.

10. In Figure 4, please insert a space between values and units. The same is valid for Figure 12.
We greatly appreciate the reviewer's suggestion. The authors have inserted spaces between values and units in Figures 4 and 2. Thanks again for the reviewer's suggestion.

11. After Figure 6, please explain item by item the corresponding function and the role developed in the system (need).
We greatly appreciate the reviewer's suggestion. The reviewer believes that the corresponding functions and the role of development in the system need to be explained item by item after Figure 6. According to the reviewer's suggestion, the author has added the following to the simulation diagram of the control system in Figure 6: According to the needs of the test, the electro-hydraulic Servo force control system realizes servo control through servo valve. The actuator of the system is a hydraulic cylinder, and one end of the hydraulic cylinder relates to the fixture. The servo mechanism is a direct-acting servo valve, and the servo valve and the hydraulic cylinder form a valve-controlled cylinder structure. The hydraulic system is powered by a plunger pump, which is connected to the motor through a coupling. In order to reduce the flow pulsation caused by the plunger pump and ensure the smooth movement of the hydraulic cylinder, an accumulator is added to the oil circuit to reduce the corresponding vibration and noise. The latest revision has been added to section 2.3 of the article, again thanks to the reviewer for the suggestion.

12. Regarding Figure 7, it is necessary to explain how a light difference between sinusoidal responses is so important.
We greatly appreciate the reviewer's suggestion. The reviewer believes that it is necessary to explain why the response between the sinusoids is so important in Figure 7. According to the reviewer's suggestion, the author is adding the following explanation: Since the pultruded sheet is tested during the test, the loading accuracy and loading frequency requirements Higher, in order to meet the actual loading requirements, the actual loading curve and the expected loading curve should ensure high synchronization, so that the slight difference between the two should be as close to zero as possible, so the slight difference between the sinusoidal responses may also affect the test results. greater impact. The latest revision has been added to section 2.3 of the article, again thanks to the reviewer for the suggestion.

13. In Figure 12 the results diverge as the load increase. Could this be avoided? Please explain how correct the simulations results.
We greatly appreciate the reviewer's suggestion. The reviewer believes that the difference between the finite element analysis and the test needs to be explained. First of all, the author adds the following explanation: In the process of numerical simulation calculation, the pultruded sheet will be considered as a uniform material, and the material parameters do not change with space. However, in the pultrusion process, the uniformity of the pultruded sheet cannot be strictly guaranteed, so the numerical simulation can only obtain a benchmark result, which cannot be strictly consistent with the test. Secondly, the author cites relevant literature to prove the difference between finite element analysis and experiment. The latest revision has been added to section 3.2 of the article, again thanks to the reviewer for the suggestion.

14. In page 14, please correct "KN" to "kN".
We greatly appreciate the reviewer's suggestion. The author has corrected "kN" to "kN" on page 14. Thanks again for the reviewer's suggestion.

15. The methodology/Approach/Strategy to develop this work doesn't exist. Thus, please try to establish and describe a strategy able to be transferred to other Researchers as Transferrable Knowledge.
We greatly appreciate the reviewer's suggestion. The reviewer believed that the word representation method could not be developed, so the author instead built a pultruded sheet tensile testing machine in the article, and did follow-up research work. The latest revisions have been added to the introduction section of the article, again thanks to the reviewers for the suggestions.

16. No discussion of the results is presented. Please take advantage of the improvements need to be made in the Introduction to include results able to be compared at final of your work.
We greatly appreciate the reviewer's suggestion. Based on the reviewer's suggestion, we have carefully revised the last paragraph of the Introduction for the study of pultruded sheets and a discussion of the results. The latest revisions have been added to the introduction to the article, again thanks to the reviewers for the suggestions.

17. The novelty and gap found in the literature remain not clear in your work.
We greatly appreciate the reviewer's suggestion. According to the reviewer's suggestion, we have carefully revised the research of many scholars in the introduction, and found the innovation of this paper, that is, constructing a testing machine to explore the tensile mechanical properties of pultruded glass fiber sheets. At the same time, there is an application-oriented innovation, which applies infrared imaging technology to the test of pultruded glass fiber sheet to discuss the damage evolution of pultruded sheet. The latest revisions have been added to the Introduction and Conclusions of the article, again thanks to the reviewers for their suggestions.

Reviewer 4 Report

The article is suitable for publication.

Author Response

We are very grateful to the reviewers for their suggestions, the authors agree with the reviewers and have revised the abstract, introduction, conclusions and other parts of the article to make the article look more organized and polished the English grammar, again Thanks to the reviewers for acknowledging the article. The reviewers feel that the English grammar of this article needs to be revised. Therefore, the authors' manuscripts have been revised by English-speaking individuals with the latest revisions added to each section of the article.

Round 3

Reviewer 3 Report

Thanks for the performed improvements.

Kind regards.